# Dual-Modality Guided Prompt for Continual Learning of Large Multimodal Models

## Abstract

Large Multimodal Models (LMMs) exhibit remarkable multi-tasking ability by learning mixed datasets jointly. However, novel tasks would be encountered sequentially in dynamic world, and continually fine-tuning LMMs often leads to performance degrades. To handle the challenges of catastrophic forgetting, existing methods leverage data replay or model expansion, both of which are not specially developed for LMMs and have their inherent limitations. In this paper, we propose a novel dual-modality guided prompt learning framework (*ModalPrompt*) tailored for multimodal continual learning to effectively learn new tasks while alleviating forgetting of previous knowledge. Concretely, we learn prototype prompts for each task and exploit efficient prompt selection for task identifiers and prompt fusion for knowledge transfer based on image-text supervision. Extensive experiments demonstrate the superiority of our approach, *e.g.*, ModalPrompt achieves **+20%** performance gain on LMMs continual learning benchmarks with **×1.42** inference speed refraining from growing training cost in proportion to the number of tasks. The code will be made publically available.

## 1 Introduction

In recent years, Large Multimodal Model (LMM), which combines visual encoder (Dosovitskiy et al., 2021) with a large language model to handle multimodal tasks, has gained remarkable performance in numerous fields including understanding and generation. As modern models become larger with billions of parameters (Dubey et al., 2024), they are expected to learn more than one time and deal with multiple tasks other than single tasks like retrieval and image caption (Yao et al., 2022; Dai et al., 2024). Typically, LMMs (Li et al., 2023; Liu et al., 2024b) apply two-stage training, first conducting multi-task pre-training on mixed datasets to establish image-text alignment and then fine-tuning on the downstream dataset to achieve superior performance.

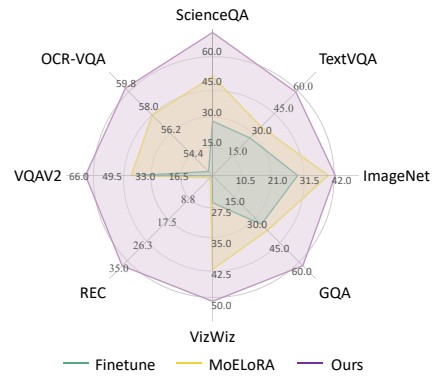

Figure 1: Performance comparison on continual learning benchmark for LMMs.

However, while pre-trained model like LLaVA (Liu et al., 2024b) performs well on mixed datasets, they tend to forget older tasks when fine-tuned on new task. Such forgetting phenomenon is especially evident in sequentially learning of widely differing multimodal tasks such as VQA (Goyal et al., 2017) and grounding (Deng et al., 2021). This calls for continual learning of multimodal large language model, which aims at sequentially fine-tuning models with multimodal tasks and gets superior performance on new tasks while remaining ability on older tasks.

Existing approaches mainly tackle the forgetting issue from two aspects. (1) Some store part of training data of older tasks and mix them with dataset of current task to resist forgetting (Rebuffi et al., 2017). However, rehearsal based method has difficulty caching data from all previous tasks and may struggle with severe issues involving data privacy and safety, especially in the era of big data, where people care more about data leakage; (2) others continually extend the model with separate lightweight components for each task, and LoRA (Hu et al., 2022) appears to be the common

practice for large models (Wang et al., 2023). However, the frequently employed model expansion methods expand model size in proportion to the number of tasks since they store separate components for each task and integrate them during inference. As the number of tasks increases, the cost of storage and inference becomes unbearable, particularly in LMMs and therefore hinder their practical deployments in real-world scenarios. Moreover, as they are not specially designed for LMMs, they often perform poorly on multimodal benchmarks. The mentioned shortcomings naturally raise an open question: *Can we establish a continual learning framework tailored for LMMs that is rehearsal-free while refraining from computational expansion in proportion to the number of tasks?*

In this paper, we investigate how to retain information of older tasks from dual-modaility (*i.e.*, image and text) and therefore improve the performance of continual learning. Generally speaking, given the suboptimal performance of the existing methodology on LMM continual learning and that the primary distinction between LLM and LMMs lies in their utilization of image features, we introduce prompt learning and build a general prompt learning framework for continual learning with supervision from multimodality. ***First***, we build a set of prototype prompts for each task to represent task-specific knowledge without the necessity of storing and replaying old samples. ***Second***, to address the problem of the increasing computational complexity associated with the growing number of tasks, we develop the prompt selection mechanism. Concretely, we use off-the-shelf text and visual encoders of CLIP (Radford et al., 2021) to obtain *text and visual guidance features*, which represent image-text distribution in feature space. At the same time, to further enhance knowledge transfer, a learnable lightweight projection layer (*e.g.*, MLP) is exploited to extract *prototype features* from prototype prompts for multi-task prompt fusion. We then obtain prototype features that are most relevant to the current task through dual-modality guidance to promote the performance.

Our method mainly has two advantages. On the one hand, features after tokenization (text) and projection (image) naturally align the dual-modality information and are effortless to retain knowledge of both modalities without data from older tasks. On the other hand, computational complexity is in proportion to the number of tokens other than the number of tasks, therefore we can manage the time consumption by selecting the number of tokens. We evaluate our approach on continual learning benchmark for LMMs (Chen et al., 2024) across diverse multi-modal tasks from VQA (Goyal et al., 2017) to grounding (Kazemzadeh et al., 2014) with various indicators. Comprehensive results certificate that our method efficiently tunes on new task, substantially boosts performance on older tasks and even achieves comparable performance to multi-task learning. Our contributions are summarized as follows:

- To the best of our knowledge, this is the first prompt learning framework for rehearsal-free continual learning of LMMs to exploit the advantage of multimodal supervision.

- We construct prototype prompts to retain knowledge from previous tasks and exploit an effective dual-modality guided prompt selection and fusion technique to manage the computational complexity and ensure continual learning ability.

- We conduct extensive experiments on large-scale continual learning benchmark for LMMs, and the results outperform existing methods by a substantial margin (**+20%**). We also give comprehensive analysis to showcase the effectiveness and efficiency of our method.

## 2 RELATED WORK

**Large Multimodal Models.** Large multimodal models (LMMs) (Liu et al., 2024b;a; Ye et al., 2024), which combine vision representation with large language models (LLMs) (Alayrac et al., 2022; Touvron et al., 2023), have exhibited prpredominant function in numerous multimodal tasks (Liu et al., 2023; Fu et al., 2023; Lu et al., 2024). They typically consist of a LLM decoder with stacks of transformers to decode textual embeddings, a vision encoder and a linear projector trained on large-scale vision-language datasets to align image-text features and project visual representations into text space. Usually, they first process image pixels with a CLIP image encoder, align features with a linear projector and then generate responses with concatenation of both image-text representations in an autoregressive way as LLMs do.

As full fine-tuning is time-consuming and resource-intensive, efficient tuning is the common practice to reduce the training cost of large models (Han et al., 2024). Methods for parameter efficient tuning

are mainly three-fold: adapter learning (Zhang et al., 2021), prompt learning (Zhou et al., 2022) and LoRA (Hu et al., 2022). They update the model with a lightweight module in the form of intra-block parallel connections, prefixes among input embeddings and low-rank decomposition, respectively. Mainstream methods employ LoRA as the solution to reduce source consumption for large models (Smith et al., 2024; Qin et al., 2024).

**Continual Learning.** Continual learning solves the problem of catastrophic forgetting (Zhai et al., 2023) when one model sequentially learns multiple tasks. Conventional works are divided into three categories: regularization based (Kirkpatrick et al., 2017; Zhu et al., 2021), rehearsal based (Rebuffi et al., 2017; Buzzega et al., 2020; Liu et al., 2021; 2020; Luo et al., 2023) and architecture based (Smith et al., 2023; Wang et al., 2022a) methods. Specifically, rehearsal based methods are effective but rely heavily on the quality of data. Architecture methods require similar model expansion that model size grows in proportion to the number of tasks, which restricts their practical application. There has been research for vision-language model (Radford et al., 2021). Specifically, L2P (Wang et al., 2022b) enhances continual learning through prompts from a memory space. Nevertheless, they do not involve large language model and all concentrate on classification tasks (Zheng et al., 2023; Zhai et al., 2023).

Since the extensive development of LLM, much attention and effort have been paid to the continual learning of LLMs (Wu et al., 2024; Zhang et al., 2024). Efficient tuning shows the potential of promoting the performance of continual learning as the backbone is usually frozen to reserve prior learned knowledge (Gao et al., 2024). Progressive Prompts (Razdaibiedina et al., 2023) assigns a set of prompts for each task and accumulates them as the number of tasks grows. Pop (Hu et al., 2023) additionally set prompt of prompts to capture cross-task information. However, they focus on NLP tasks (Wang et al., 2024b;a; 2023) with no special design for visual features and few works explore continual learning of LMMs (He et al., 2023). CoIN (Chen et al., 2024) proposes a multimodal continual learning benchmark and applies MoELoRA (Dou et al., 2023) to align previous instructions. However, it suffers from severe performance drop, indicating that LoRA might not be the final solution to multimodal continual learning. In this paper, we focus on continual learning for multimodal tasks and construct prompt learning scheme tailored for LMM continual learning and computational consumption.

# 3 MODALPROMPT: A NOVEL PROMPT BASED CONTINUAL LEARNING FRAMEWORK FOR LMMS

Continual learning of LMMs seeks to address the issue of learning with ongoing datasets. Denote that LMM $f_\theta(\cdot)$ is pre-trained on large-scale vision-language data to align image-text features. Given $T$ tasks $\{\mathcal{T}_1, \cdots, \mathcal{T}_T\}$ with corresponding multimodal data $\mathcal{D}_t = \{X_v^{t,i}, X_{instruct}^{t,i}, y^{t,i}\}_{i=1}^{N_t}, t = 1, \cdots, T$, where $X_v^{t,i}, X_{instruct}^{t,i}, y^{t,i}$ stands for $i^{th}$ sample of image, text and ground truth for $t^{th}$ dataset ($N_t$ in total). A continual learner aims to fine-tune $f_\theta(\cdot)$ sequentially on current data $D_t$ while retaining knowledge on all previous tasks $\mathcal{T}_{<t}$. For a given dataset $D_t$, multimodal model generates responses for each input $\{X_v^t, X_{instruct}^t\}$ after aligning and concatenating image-text features:

$$f([X_v^t; X_{instruct}^t]; \theta_t), \tag{1}$$

where $[\cdot; \cdot]$ represents concatenation operation. Fine-tuning objective for LMMs is a negative log-likelihood auto-regressive language loss and when learned continuously, the model is sequentially optimized on different tasks and $\theta$ is continuously updated to adapt to newly emerged dataset:

$$\mathcal{L}_{LMM}(\theta) = \mathbb{E}_{(X_v^t, X_{instruct}^t, y) \sim \mathcal{D}_t} \left[ - \sum_{\ell=1}^{L} \log p_\theta(y^\ell | X_v^t, X_{instruct}^t, y^{<\ell}) \right], \tag{2}$$

where $L$ is the length of each sample pair in the dataset. The model predicts the answer in an auto-regressive way, *i.e.*, outputs the $\ell^{th}$ response conditioned on all instruction and answer tokens before index $\ell$. Sequentially learning continuous tasks will do favor for new tasks, but may cause catastrophic forgetting in older tasks.

In this paper, we aim to resolve the problem of continual learning in a more challenging setting. The characteristic of LMM continual learning includes: (1) **diverse multimodal generative questions:** continual learning procedure is focused on generative tasks other than discriminative tasks

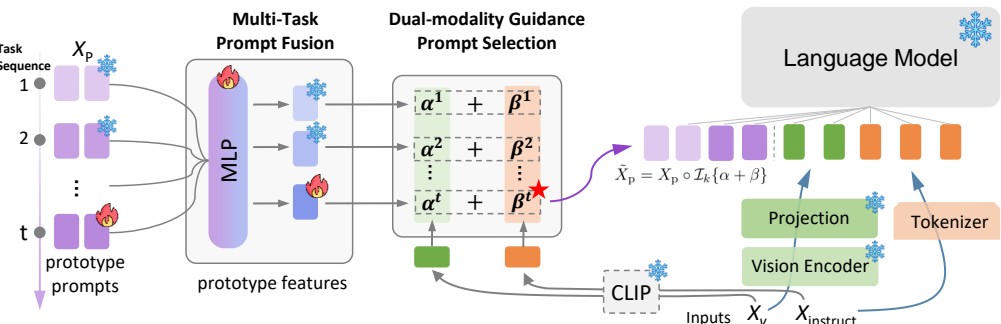

Figure 2: *Left:* prompt selection module. Prototype features are obtained from the projection of prototype prompts to get task-specific knowledge in feature space. *Middle:* dual-modality guidance process. Prototype features that are the most similar to current multimodal features are selected to enhance training and evaluated tasks. *Right:* prototype prompts and original multimodal inputs are concatenated and fed into large language model to generate responses.

like image classification (Wang et al., 2022b) and with the existence of vision information, type of task is much more diverse and covers abundant scenarios; (2) **free from task identifiers:** during inference, the model does not possess prior knowledge regarding which specific task current question belongs to; (3) **absence of replay samples:** due to data privacy, no samples are replayed to refresh knowledge of previous tasks.

## 3.1 DUAL-MODALITY GUIDANCE PROMPT SELECTION FOR TASK IDENTIFIERS

We start from direct fine-tuning of LMMs employing prompt learning framework, *i.e.*, tuning independent prompts for respective datasets. Given a set of prompts $X_{\mathrm{p}}^t$ with length $N$ that is well-trained on task $\mathcal{T}_t, t \in \{1, \cdots, T\}$ in the form of direct fine-tuning, the crucial problem is that the model has no ability to recognize which set of prompts promotes particular datasets during inference, *i.e.*, without access to data from older tasks, task-specific prompts should obtain cues for image-text distribution and be discriminant about which set of prompts counts during inference. Therefore, it is necessary to measure similarity between image-text distribution of certain tasks and task-specific prompts. To achieve this goal, we propose the dual-modality guidance for prompt selection during evaluation to tackle the issue. Specifically, for representations of each set of prompts, we use the average of prompts as prompt features:

$$\boldsymbol{x}_{\mathrm{p}}^t = \frac{1}{N} \sum X_{\mathrm{p}}^t. \tag{3}$$

Considering that CLIP well captures image-text distributions in features space, for image $X_{\mathrm{v}}$ and text $X_{\mathrm{instruct}}$ in each sample of current task (without identifiers), we reuse off-the-shelf vision and text encoder from CLIP to extract multimodal knowledge of specific task:

$$\boldsymbol{x}_{\mathrm{v}} = \mathrm{Proj}_{\mathrm{v}}(E_I(X_{\mathrm{v}})), \quad \boldsymbol{x}_{\mathrm{instruct}} = E_T(X_{\mathrm{instruct}}), \tag{4}$$

where $E_I(\cdot) : \mathbb{R}^{n_{\mathrm{v}} \times d_{\mathrm{v}}} \to \mathbb{R}^{d_{\mathrm{v}}}, E_T(\cdot) : \mathbb{R}^{n_{\mathrm{t}} \times d_{\mathrm{t}}} \to \mathbb{R}^{d_{\mathrm{t}}}$ and $\mathrm{Proj}_{\mathrm{v}}(\cdot) : \mathbb{R}^{d_{\mathrm{t}}} \to \mathbb{R}^{d_{\mathrm{v}}}$, are CLIP vision encoder, text encoder and linear projection, respectively. $n_{\mathrm{v}}, n_{\mathrm{t}}, d_{\mathrm{v}}$ and $d_{\mathrm{t}}$ are length of image inputs, length of text inputs, visual dimension and textual dimension, respectively. The utilization can be effortless as they are well-trained and frozen for feature extraction.[1]

The dual-modality features could serve as guiding cues for selecting prompts that are close to multimodal distributions of current task in feature space. Concretely, we exploit the similarity of prompt features with dual-modality features, respectively:

$$\alpha^t = \mathrm{sim}(\boldsymbol{x}_{\mathrm{p}}^t, \boldsymbol{x}_{\mathrm{v}}), \quad \beta^t = \mathrm{sim}(\boldsymbol{x}_{\mathrm{p}}^t, \boldsymbol{x}_{\mathrm{instruct}}), \ t = 1, \cdots, T, \tag{5}$$

where similarity is a measurement that defines the correlation between features, and we use simple yet effective cosine similarity as:

$$\mathrm{sim}(\boldsymbol{x}_i, \boldsymbol{x}_j) = \frac{\boldsymbol{x}_i \cdot \boldsymbol{x}_j}{||\boldsymbol{x}_i|| \ ||\boldsymbol{x}_j||}. \tag{6}$$

---

[1] As LMM uses vision encoder to extract image features, extra consumption merely comes from text encoder.

With dual-modality guidance, the model has the ability to determine which prompts may boost the performance of evaluated task. We then select the prompts among $1, \cdots, T$ with the largest similarity of multimodal supervision:

$$\tilde{X}_{\mathrm{p}} = X_{\mathrm{p}} \circ \mathcal{I}_k\{\alpha + \beta\}, \tag{7}$$

where $\mathcal{I}_k$ represents selecting the index with the largest $k$ elements, and $\circ$ means selecting according to index. The dual-modality guidance prompt selection module has to advantages: (1) help choose the tasks which may help boost the performance; (2) manage the inference speed as the time complexity are in proportion to the number of selected prompts other than the number of tasks.

**Response generation.** For each evaluated task, prompt learning feeds several efficient prompts together with multimodal inputs in a prefix way to generate answers for multimodal inputs:

$$f([\tilde{X}_{\mathrm{p}}; X_{\mathrm{v}}; X_{\mathrm{instruct}}]; \theta), \tag{8}$$

where $\tilde{X}_{\mathrm{p}}$ is the selected prompts through prompt selection module.

### 3.2 Multi-Task Prompt Fusion for Knowledge Transfer

Another key issue for continual learning is how to retain knowledge from older tasks and boost the performance of current task. Motivated by the dual-modality prompt selection, we propose to transfer similar knowledge in training procedure through multi-task prompt fusion, in which we continually integrate knowledge of older tasks during sequential prompt learning. We term set of prompts for each task as prototype prompts. The difference lies in that during training, prototype prompts of all previous tasks are frozen for knowledge reuse and only current prototype prompts are trainable, as shown in Fig. 2.

When training the $t^{th}$ task, the trainable prototype prompts are supposed to draw close to vision-language features of current task and absorb potential knowledge that may boost the performance. To enhance knowledge transfer, the dual-modality features could serve as guiding cues for prompts to accurately get close to multimodal distributions of current task in feature space. Therefore, we build prototype features from a lightweight projection layer:

$$\boldsymbol{x}_{\mathrm{p}}^t = \mathrm{Proj}_{\mathrm{p}}(X_{\mathrm{p}}^t), \tag{9}$$

where $\mathrm{Proj}_{\mathrm{p}}(\cdot): \mathbb{R}^{N \times d_t} \to \mathbb{R}^{d_t}$ projects the prototype prompts into task-specific prototype features in image-text feature space. It is effective in distinguishing whether prompts of older tasks are favorable for current tasks, *i.e.*, fusing prompts of similar tasks would enhance knowledge transfer and consequently boost the performance.

To explicitly utilize the knowledge of prior tasks, we design multi-task prompt fusion to figure out prototype prompts that promote current task. Concretely, we fuse the prototype prompts among $1, \cdots, t$ with the largest similarity of multimodal supervision for knowledge transfer:

$$\tilde{X}_{\mathrm{p}}^t = X_{\mathrm{p}}^{\leq t} \circ \mathcal{I}_k\{\alpha^{\leq t} + \beta^{\leq t}\}. \tag{10}$$

It differs from Eqn. 7 in that during training, only prompts trained on older tasks can be selected. In order to optimize parameters of current task, prototype prompts of current task are always selected.

Intuitively, we explicitly integrate prompt fusion into training procedure and utilize supervision from both modalities that caters for LMMs to measure the distance with distribution of current task and therefore transfer previous knowledge to boost the performance of current task, *i.e.*, prototype prompts that are close to current feature distribution. Detailed analyses are shown in Sec. 4.2.

**Training objectives.** Different from evaluation process, the inputs for continual learning of task $\mathcal{T}_t$ are prefixed with fused prototype prompts $\tilde{X}_{\mathrm{p}}^t$ described above. The parameters of large language model $\theta$ are frozen, and only parameters of prototype prompts corresponding to current task $\theta_{\mathrm{p}}^t$ are trainable. The optimization target for task $\mathcal{T}_t$ is to find optimal parameters $\theta_{\mathrm{p}}^t$ that minimize the language loss:

$$\mathcal{L}_{\mathrm{LMM}}^t(\theta_{\mathrm{p}}^t) = \mathbb{E}_{(X_{\mathrm{v}}^t, X_{\mathrm{instruct}}^t, y^t) \sim \mathcal{D}_t} \left[ -\sum_{\ell=1}^{L} \log p(y^\ell | [\tilde{X}_{\mathrm{p}}^t, X_{\mathrm{v}}, X_{\mathrm{instruct}}, y^{<\ell}], \theta, \theta_p^1, \cdots, \theta_p^t) \right]. \tag{11}$$

Additionally, the projection layer along with prototype prompts of current task is optimized together to reserve prototype feature during training process. Since we are to *maximum* the similarity with dual-modality features to keep knowledge of current task, we design prototype loss as:

$$\mathcal{L}^t_{\text{Proto}} = \left[1 - \text{sim}(\boldsymbol{x}^t_{\text{p}}, \boldsymbol{x}_{\text{instruct}})\right] + \left[1 - \text{sim}(\boldsymbol{x}^t_{\text{p}}, \boldsymbol{x}_{\text{instruct}})\right]. \tag{12}$$

Total training objective is the sum of the prototype similarity loss and language loss:

$$\mathcal{L}^t_{\text{Total}} = \mathcal{L}^t_{\text{Proto}} + \mathcal{L}^t_{\text{LMM}}. \tag{13}$$

The trainable parameters are optimized with both understanding responses and learning prototypes in feature spaces in the training procedure. Parameters of current task are frozen afterwards and are used to retain knowledge of learned tasks when new task occurs.

# 4 EXPERIMENTS

## 4.1 SETUP

We apply LLaVA (Liu et al., 2024b) as base LMM, and CLIP-Large-336 (Radford et al., 2021) as vision and text encoder for dual-modality feature extraction. The prototype prompts can be easily constructed by extending the vocabulary size of the language tokenizer. The length for each prototype prompt is set to 10. We select 3 prototypes for LMM prompt learning. Implementation details are shown in Appendix A.2.

**Datasets.** We follow the setting of CoIN (Chen et al., 2024), which is a continual instruction tuning benchmark for LMMs, and employs numerous vision-language tasks to evaluate the continual learning ability. Datasets are composed of GQA (Hudson & Manning, 2019), OCRVQA (Mishra et al., 2019), Vizwiz (Gurari et al., 2018), VQAv2 (Goyal et al., 2017), ScienceQA (Lu et al., 2022), TextVQA (Singh et al., 2019), ImageNet (Deng et al., 2009) and RefCoco (Mao et al., 2016; Kazemzadeh et al., 2014). Most of these datasets are visual question answering tasks of different fields, *e.g.*, GQA for visual reasoning and ScienceQA for science knowledge, except for ImageNet (classification) and RefCoco (grounding). More details about instructions and statistics can be found in Chen et al. (2024).

**Evaluation metrics.** Denote that $A_{t,i}(i \leq t)$ is the performance of task $i$ after training on task $t$ ($T$ tasks in total). For final performance evaluation (number of dataset as the variable for a given incremental stage), we measure each dataset using metrics like **DirectT** (directly fine-tuning the initial LMM with each data, *i.e.*, $A_{i,i}, i = 1, \cdots, T$, which solely focuses on the effectiveness of fine-tuned task) and **ContinualT** (evaluating after sequential training on all tasks, *i.e.*, $A_{T,i}, i = 1, \cdots, T$). For time-dependent continuous evaluation (number of incremental stage as the variable for given datasets), we evaluate continuous metrics at each incremental stage across all seen datasets. Other metrics include:

(1) Backward Transfer (BWT): $B_t = \frac{1}{t-1} \sum_{i=1}^{t-1} (A_{i,i} - A_{t,i}), t = 2, \cdots, T$. It reflects the relative variation between current performance and direct tuning performance, measuring the catastrophic forgetting on all tasks. Lower BWT represents better anti-catastrophic forgetting performance.

(2) Mean Accuracy (MA): $M_t = \frac{1}{t} \sum_{i=1}^{t} A_{t,i}$. It measures the average performance of all tasks at each incremental stage and is introduced to evaluate continual learning ability of all previous tasks. Higher MA stands for better continual learning ability. The above two metrics are averaged across all data on each incremental stage except the first one, *i.e.*, $t = 2, \ldots, T$.

(3) Continual Average Accuracy (CAA): In addition to ContinualT, which focuses on performance after tuning on all datasets, we propose to average performance throughout the entire tuning process. $CAA_i = \frac{1}{T-i} \sum_{t=i+1}^{T} A_{t,i}, i = 1, 2, \ldots, T-1$. It measures the absolute performance of each data across the sequential tuning. It is vital to keep the performance from dropping severely when the fine-tuning task varies greatly.

Table 1: Comprehensive comparison of continual learning ability. DirectT is instantly evaluated after tuning on corresponding dataset and ContinualT is evaluated after tuning on OCR-VQA.

| Metric | Method | ScienceQA | TextVQA | ImageNet | GQA | VizWiz | REC | VQAV2 | OCRVQA |
|--------|--------|-----------|---------|----------|-----|--------|-----|-------|--------|
| | Multi-task | 46.22 | 47.19 | 95.47 | 56.40 | 53.35 | 34.27 | 58.62 | 55.08 |
| | Zero-shot | 49.91 | 3.31 | 2.17 | 3.02 | 0.85 | 0.00 | 0.68 | 1.05 |
| DirectT | Finetune | 82.45 | 50.14 | 95.01 | 55.65 | 51.42 | 34.00 | 59.17 | 52.92 |
| | MoELoRA | 75.78 | 51.80 | 79.60 | 57.95 | 58.70 | 36.77 | 64.58 | 57.50 |
| | Ours | 77.05 | 58.50 | 42.26 | 62.17 | 48.81 | 36.88 | 66.91 | 59.68 |
| ContinualT | Finetune | 26.00 | 25.38 | 28.51 | 33.07 | 26.52 | 0.10 | 40.00 | 52.92 |
| | MoELoRA | 47.34 | 32.91 | 38.73 | 37.15 | 42.48 | 0.97 | 42.77 | 57.50 |
| | Ours | **68.42** | **56.40** | **41.13** | **61.11** | **50.13** | **36.69** | **66.90** | **59.68** |
| | Δ | +21.08 | +23.49 | +2.40 | +23.96 | +7.65 | +35.72 | +24.13 | +2.18 |
| CAA | Finetune | 13.79 | 15.74 | 17.30 | 28.84 | 15.20 | 0.06 | 40.00 | - |
| | MoELoRA | 39.12 | 27.10 | 20.01 | 40.65 | 28.72 | 1.36 | 42.77 | - |
| | Ours | **68.36** | **56.30** | **39.66** | **61.45** | **50.02** | **36.66** | **66.90** | - |
| | Δ | +29.23 | +29.20 | +19.65 | +20.80 | +21.30 | +35.30 | +24.13 | - |

Table 2: Continual performance metrics at each incremental stage.

| Method | TextVQA | | ImageNet | | GQA | | VizWiz | | REC | | VQAV2 | | OCRVQA | |
|--------|---------|---------|----------|---------|-----|-----|--------|-----|-----|-----|-------|-----|--------|-----|
| | $B_2 \downarrow$ | $M_2 \uparrow$ | $B_3 \downarrow$ | $M_3 \uparrow$ | $B_4 \downarrow$ | $M_4 \uparrow$ | $B_5 \downarrow$ | $M_5 \uparrow$ | $B_6 \downarrow$ | $M_6 \uparrow$ | $B_7 \downarrow$ | $M_7 \uparrow$ | $B_8 \downarrow$ | $M_8 \uparrow$ |
| Finetune | 44.30 | 44.14 | 65.53 | 32.18 | 52.62 | 31.35 | 51.43 | 25.79 | 66.16 | 6.31 | 43.40 | 23.92 | 35.47 | 29.06 |
| MoELoRA | 41.31 | 43.13 | 52.47 | 34.08 | 32.76 | 41.71 | 33.81 | 37.71 | 41.41 | 25.59 | 30.80 | 34.34 | 26.12 | 37.48 |
| Ours | **6.55** | **64.50** | **4.40** | **56.34** | **3.16** | **57.63** | **4.51** | **54.15** | **3.98** | **50.96** | **2.02** | **54.35** | **1.68** | **55.06** |

We employ ContinualT, CAA and BWT, MA to measure the performance of final continual performance and continuous continual performance, respectively.

## 4.2 MAIN RESULTS

**Final continual performance.** We sequentially fine-tune data from benchmark in the order of ScienceQA, TextVQA, ImageNet, GQA, VizWiz, REC, VQAV2 and OCRVQA and evaluate after tuning on all tasks. From Tab. 1, we can conclude that: **(1)** Our method achieves remarkable improvements on all continual learning metrics (ContinualT and CAA), and substantially outperforms existing methods with **+20%** gain (**+17.6%** and **+25.7%** for ContinualT and CAA, respectively). Notably, the results after sequential tuning (ContinualT) even against multi-task training, strongly demonstrating the effectiveness of the dual-modality guided prompt learning framework. **(2)** When learning different types of tasks, our approach undergoes slight performance drop and still gets competitive results other than losing the ability to respond to the task (decreasing to zero when MoELoRA is evaluated on Grounding), indicating the continual learning ability of the proposed method. **(3)** CAA of previous methods drop significantly compared with ContinualT, indicating that CAA more comprehensively reflects continuous learning performance, and CAA of our method has **almost no degradation**, implying that our method consistently achieves superior performance across the continuous tuning. **(4)** Tasks equipped with our task selection module are able to benefit from similar tasks, *e.g.*, ImageNet/Grounding, VQA tasks, which is favourable for inter-task boosting when number of tasks are increasing. This strongly certificates that our prompt fusion and selection module is helpful in retaining knowledge of previous tasks and promoting the performance of similar tasks. See Appendix A.1 for full experimental results.

**Continuous continual performance.** We also evaluate continuous metrics at each incremental stage in Tab. 2 to examine the time-variant continual learning performance. In particular, compared with previous methods, our method is especially effective in alleviating catastrophic forgetting (BWT) to the most (**33.2% mitigation**) and also gets promotion in absolute performance evaluation (**19.9%** concerning MA). It is evident that we outperform the state-of-the-art LoRA-base method by a substantial margin with respect to both anti-catastrophic forgetting and enhancing mean accuracy, further validating the superiority of our approach.

Table 3: Effectiveness of guidance from multimodal supervision. Dual-modality similarity guidance achieves the best results.

| Guidance | ScienceQA | TextVQA | ImageNet | GQA | VizWiz | REC | VQAV2 | OCRVQA |
|---|---|---|---|---|---|---|---|---|
| Only Image | 66.24 | **56.94** | 21.49 | 60.46 | 49.98 | 36.36 | 66.55 | 57.59 |
| Only Text | 65.59 | 55.90 | 13.41 | 60.65 | 47.86 | 36.18 | 66.33 | 57.21 |
| Dual Modality | **68.42** | 56.40 | **41.13** | **61.11** | **50.13** | **36.69** | **66.90** | **59.68** |

Table 4: Effectiveness of the proposed prompt selection and fusion for continual learning. Both of them plays a key role in the framework and lacking either of them causes severe performance drop.

| fusion | selection | ScienceQA | TextVQA | ImageNet | GQA | VizWiz | REC | VQAV2 | OCRVQA |
|---|---|---|---|---|---|---|---|---|---|
|  | ✓ | 44.28 | 50.36 | 34.80 | 43.56 | 46.28 | 7.00 | 37.71 | 34.90 |
| ✓ |  | 52.53 | 52.26 | 37.02 | 51.70 | 47.35 | 10.37 | 54.26 | 53.02 |
| ✓ | ✓ | **68.42** | **56.40** | **41.13** | **61.11** | **50.13** | **36.69** | **66.90** | **59.68** |

## 4.3 ABLATION STUDY

We conduct numerous ablation studies to carefully validate the effectiveness of components and hyper-parameters in the proposed method.

**Effectiveness of dual-modality guidance.** The dual-modality guided prompt selection is the core component of the proposed prompt learning framework. The difference between LMM continual learning and that of large language model mainly lies in the information incorporated from image features. Therefore, we comprehensively consider exploiting multimodal supervision using a mixture of dual-modality guidance. To this end, we analyze the impact of dual-modality guidance and replace it with single-modality guidance.

It is evident in Tab. 3 that either image or text information solely performs inferior to the proposed multimodal strategy, and image information from multimodal dataset plays an inescapable function in guiding continual learning especially in datasets that rely heavily on image scenes like TextVQA. This strongly showcases that our dual-modality guidance tailored for LMMs suits the best and largely improves the performance of multimodal continual instruction tuning by retaining robust and reliable prototype features in feature space and therefore helping multimodal features from all continuous tasks.

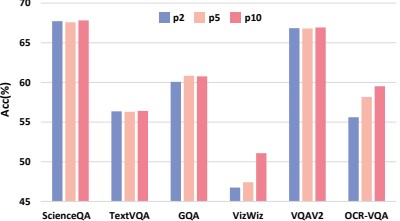 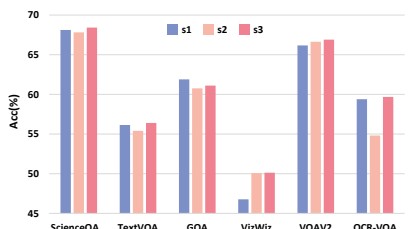

Figure 3: Impact of number of prototype prompts. p2/p5/p10 represents 2/5/10 number of prototype prompts for each task, respectively.

Figure 4: Influence of number of prompt selection. s1/s2/s3 stands for selecting 1/2/3 number of prompts during evaluation, respectively.

**Effectiveness of prompt selection and fusion.** We design the dual-modality prompt selection for task identifier and multi-task prompt fusion for knowledge transfer. To validate the effectiveness of the proposed mechanisms, we ablate each of them to demonstrate their usefulness. Specifically, without prompt selection, we concatenate all prompts like Progressive Prompts (Razdaibiedina et al., 2023). It is shown in Tab.4 that multi-task prompt fusion is significant in promoting the continual learning in the form of knowledge transfer. Also, without selection, knowledge of different types of tasks would confuse the model and lead to performance drop.

**Prototype prompts.** The number of prototype prompts represents prototype features in aligned image-text space and the similarity between feature distribution of current multimodal data plays a key role in performance stability and robustness of continual learning. Drawing on the parameter

Table 5: Efficiency comparison of LoRA based methods (MoELoRA) and ours. We average the training time for one epoch across all datasets.

| Method | GPU memory (Model)↓ | GPU memory (Total)↓ | Training time↓ | Inference time↑ | Trainable parameters↑ |
|--------|---------------------|---------------------|----------------|-----------------|-----------------------|
| MoELoRA | 15564 $M$ | 16784 $M$ | 10.74 $h$ | 2.41 token/$s$ | 4.73% |
| Ours | **14055 $M$** | **15517 $M$** | **3.81 $h$** | **3.43 token/$s$** | **0.27%** |

selection of prompt based methods in LLM (Razdaibiedina et al., 2023), we set the number of prototype prompts for each task to 10. We alternate the number of prototype prompts to analyze its stability. Results in Fig. 3 elucidate that increasing prompt numbers brings slight performance improvement. Considering both effectiveness and efficiency, we do not expand the quantity.

**Selection features.** One primary advantage of the prompt learning framework is that it relies on the number of prefix prompts other than task numbers and we can keep the number of prompts unchanged through prompt selection strategy. In this section, we explore the influence as the number of selection prompts $k$ varies in Fig. 4. It is illustrated that the performance is generally proportional to the number of selection features with diminishing marginal benefits. It is also notable that selecting one set of prompts performs better than employing two sets, and three yields the best results. It can be interpreted that one set focuses on exploiting prompts of current tasks only, and produces slightly better outcomes than two sets, in which features between previous and current tasks may disturb the representation and generate suboptimal results. However, one set does not consider facilitation among similar tasks and still gets inferior results. By contrast, three sets bring performance gain, which indicates that aggregating more prompts of similar feature distribution is consistent with the objective of continual learning and as a result boosts the performance.

Taking both proficiency and efficiency into account, we choose three sets of prototype prompts, *i.e.*, 30 prompts in total to concentrate on both retaining knowledge of older tasks and reducing computational complexity.

### 4.4 FURTHER ANALYSIS

**Efficiency comparison.** As prompt learning serves as another way to efficiently fine-tune large models, it is necessary to assess the efficiency of the methods. Therefore, we compare our approach with LoRA based method (Chen et al., 2024) in terms of additional parameters, average inference latency and GPU memory consumption. Tab. 5 reveals that our strategy achieves better results with lower memory, lower inference latency and lower trainable parameters. Specifically, we merely train **0.27%** of total parameters, which is **5%** of MoELoRA. Therefore, our method achieves faster inference speed (×**1.42**), reduces training time (×**0.35**) and GPU memory consumption, firmly substantiating the efficiency of our approach. The achievements can be attributed to simple prompt learning implementation and the prompt selection module that manages the computational complexity, consequently improving the inference efficiency.

**Similarity of dual-modality features.** The ability of our framework to learn continually is largely guaranteed by the prompt selection module and prototype prompts represented in vision-language feature space. To further analyze the effectiveness of the dual-modality guidance tailored for LMMs, we calculate the similarity matrix between prototype prompts and multimodal task guidance. In Fig. 5, the similarity heatmap vividly illustrates the vision-language distributions of continual learning tasks. First, multimodal features of a few tasks are similar (reflected by mostly large values in the similarity matrix), showcasing that most multimodal tasks share common sense and can promote each other continually. However, some tasks, such as GQA and OCRVQA, are not similar to other tasks, which may be due to their task-specific ability not needed by other common tasks (visual reasoning for GQA and OCR for OCRVQA);



Figure 5: Similarity between prototype features (column) and multimodal task features (row). Larger value indicates more similar distribution.

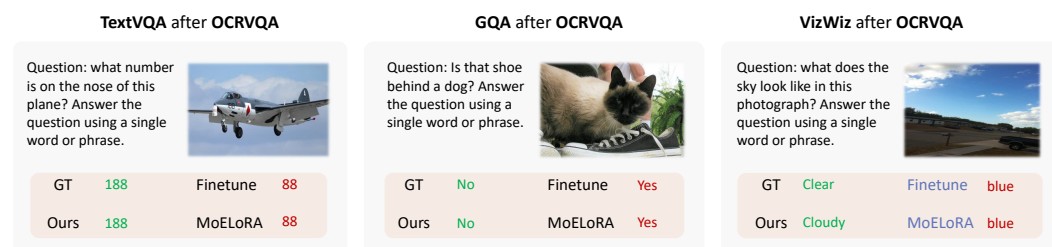

Figure 7: Continual learning responses of several examples from TextVQA, GQA and VizWiz after fine-tuning on OCRVQA. Our method can maintain the performance of previous tasks.

second, the similarity is asymmetric, which may be attributed to their task inclusion relationship. For instance, GQA requires higher-level reasoning ability, while some other tasks may merely need to answer questions based on visual-language information. Therefore, features of GQA task are similar to the prototype features of other tasks (more specifically), but other tasks are not similar to the prototype of GQA (more basically). The visualization of dual-modality features exhibits the connection between prior obtained knowledge (prototype features) and given task (multimodal task features), and therefore contributes fundamentally to continual learning ability of LLMs.

**Selection of prototype prompts.** To figure out the actual selection of prototype prompts during inference in addition to soft distribution construction and help have an intuitive understanding of the function of prompt selection module, we report the selection results of each previous task in percentage under continual learning setting, *i.e.*, ContinualT, in Fig. 6. The results expose that the proposed module correctly matches and prioritizes prototype prompts of the corresponding task as prefixes to enhance the continual learning performance of LLMs, demonstrating the robustness and usefulness of the learned prototype features. Additionally, the module also selects prototype prompts from tasks of similar type, which similarly achieves excellent performance. This strongly indicates that tasks of the same type can mutually promote the performance, and our method leverages this characteristic excellently.

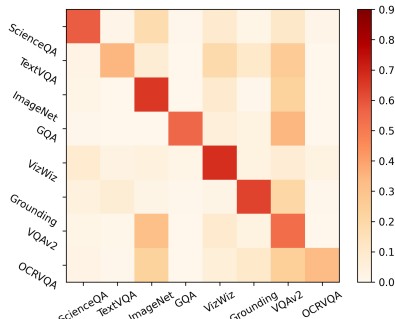

Figure 6: Selection probability of each task (row) from prototype prompts (column). Results are reported in percentage so the sum of rows equals one.

**Visualization.** Fig. 7 provides examples during continual learning procedure to explicitly illustrate the effectiveness of our method. It is elucidated in Fig. 7 that our method can maintain performance on different types of previous tasks. Concretely, our model keeps general knowledge and the capability to answer the question requiring comprehensive understanding. For example, the model identifies the specific part location of objects (nose of the plane), overcomes occlusion in TextVQA and distinguishes spatial orientation, identifies objects in GQA. Moreover, it also deduces appropriate answers with analogous meanings to the ground truth based on image and text questions (*e.g.*, cloudy and clear in VizWiz). Based on the retained knowledge, the model gives the correct answer outperforming existing continual learning methods. The visualizations strongly demonstrate that our model can retain the ability to understand and respond to diverse generative tasks instead of merely overfitting given data (output the exact ground truth) when learning continually.

## 5    CONCLUSION

In this paper, we analyze the limitations of current methods of continual learning for LMMs, and propose to exploit prompt learning for continually learning image-text generative tasks to retain knowledge of older tasks from multimodal supervision. Specifically, we construct a set of prototype prompts for each task to represent distribution in feature space and propose prompt fusion and selection module to both enhance the performance from mutual promotion of similar tasks and manage the computational complexity of the model. Comprehensive experiments and analyses validate the effectiveness and efficiency of our framework that our model achieves substantial improvements while maintaining computational complexity.

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

# A  APPENDIX

## A.1  DETAILED CONTINUAL LEARNING RESULTS

We showcase brief results in the main results. We provide detailed continual learning performance during evaluation at each incremental stage. Upper and bottom of Tab. 6 are comparison of CoIN and Ours. It can be concluded that our method achieves consistent and significant imporvements against previous LoRA based method, validating the effectiveness of our method.

Table 6: Detail continual learning results of CoIN and our method.

| CoIN | ScienceQA | TextVQA | ImageNet | GQA | VizWiz | REC | VQAV2 | OCRVQA |
|------|-----------|---------|----------|-----|--------|-----|-------|--------|
| ScienceQA | 75.78 | | | | | | | |
| TextVQA | 34.47 | 51.80 | | | | | | |
| ImageNet | 22.61 | 0.04 | 79.60 | | | | | |
| GQA | 32.37 | 34.04 | 42.48 | 57.95 | | | | |
| VizWiz | 45.32 | 38.13 | 2.63 | 43.80 | 58.70 | | | |
| REC | 58.76 | 9.08 | 5.64 | 31.87 | 11.45 | 36.77 | | |
| VQAV2 | 33.01 | 48.42 | 10.61 | 49.78 | 32.23 | 1.75 | 64.58 | |
| OCRVQA | 47.34 | 32.91 | 38.73 | 37.15 | 42.48 | 0.97 | 42.77 | 57.50 |
| **Ours** | ScienceQA | TextVQA | ImageNet | GQA | VizWiz | REC | VQAV2 | OCRVQA |
| ScienceQA | 77.05 | | | | | | | |
| TextVQA | 70.50 | 58.50 | | | | | | |
| ImageNet | 68.57 | 58.18 | 42.26 | | | | | |
| GQA | 68.82 | 56.08 | 43.43 | 62.17 | | | | |
| VizWiz | 67.48 | 55.05 | 37.60 | 61.81 | 48.81 | | | |
| REC | 66.58 | 55.68 | 35.92 | 61.95 | 48.74 | 36.88 | | |
| VQAV2 | 68.12 | 56.43 | 40.22 | 60.92 | 51.19 | 36.63 | 64.99 | |
| OCRVQA | 68.42 | 56.40 | 41.13 | 61.11 | 50.13 | 36.69 | 65.02 | 57.59 |

## A.2  ADDITIONAL IMPLEMENTATION DETAILS

Our framework is constructed depending on deepspeed repository [2] and the instructions are from CoIN [3]. In evaluation of ImageNet, we give option choices for each question-answer pairs to avoid inaccurate descriptions. All training and evaluation experiments are conducted on NVIDIA A6000. During training, batch size is adaptively adjusted to maximize the memory utilization.

## A.3  LIMITATIONS AND FUTURE WORKS

While our method achieves substantial improvements, we conduct experiments on LLaVA-7b and does not scale the experiments due to resource limitations. We believe that the effectiveness of our framwork and will treat scaling model size and application to other LMM models as future work.

---

[2]https://github.com/microsoft/DeepSpeed
[3]https://github.com/zackschen/CoIN

