# OpenReview forum: "Dual-Modality Guided Prompt for Continual Learning of Large Multimodal Models"
_ICLR.cc/2025/Conference — ICLR 2025 Conference Withdrawn Submission_

### Official Review · Reviewer_d96g · 2024-10-27

**Soundness:** 3
**Presentation:** 1
**Contribution:** 2
**Rating:** 3
**Confidence:** 4

**Summary:**

This paper explores continual learning in Large Multimodal Models, focusing on the challenge of enabling models to continuously learn across sequential tasks. The authors critically assess the limitations of existing approaches and propose a novel dual-modality guided prompt learning framework for multimodal continual learning. Extensive experiments show that the proposed method significantly enhances both performance and inference speed.

**Strengths:**

1.  The question investigated in this paper is critical and significant in the current deep learning community.
2. The paper proposes a novel prompt learning framework for rehearsal-free continual learning of LMMs.
3. They conduct extensive experiments to demonstrate the effectiveness and inference speed of proposed methods.

**Weaknesses:**

1. Although the experiment improves the performance and inference speed, the proposed method involves modality-specific prompts for each task, which is too simple compared to existing work that devises advanced prompt strategies in visual scenarios. Simultaneously, they lack of comparison with the amount of prompt-based methods. Such as: DualPrompt [1],  L2P[2], CODA-Prompt[3].
2. There exist some typos in the paper:
     1. in line 100, `prpredominant'.
      2. in line 128, ... set `prompt of prompts' ...
3. The author proposes the setting of refrains from computation expansion in proportion to the number of tasks. Whether means we can continuously learn the sequential data in one model and the performance will continuously improve. In other words,  how many tasks can the proposed method effectively handle within one model?
4. In the experiment, there is a lack of results that compare one task in the continuous process, i.e. compare the performance at the time axes, which directly reflects the transfer capability when more previous knowledge is learned.
5. There is no difference between the two items in equation 12 with the add operation.
6. How does the proposed method assess forgetting? Does it require saving a lightweight projection layer for each task, or should the projection layer from a previous task be re-tunned after learning a new one?
7. In Line 203, why does the encoder of visual E_I and textual E_T in CLIP realize the mapping of  E_I(·) : R^{n_v ×d_v} →R^{d_v} ,E_T(·) : R^{n_t ×d_t} →R^{d^t}, which should exist error description?

[1]. DualPrompt: Complementary Prompting for Rehearsal-free Continual Learning

[2]. Learning to Prompt for Continual Learning

[3]. CODA-Prompt: COntinual Decomposed Attention-based Prompting for Rehearsal-Free Continual Learning

**Questions:**

See weakness.

---

### Official Review · Reviewer_JJqr · 2024-10-28

**Soundness:** 2
**Presentation:** 3
**Contribution:** 2
**Rating:** 3
**Confidence:** 3

**Summary:**

This paper presents MODALPROMPT, a dual-modality guided prompt framework designed to address catastrophic forgetting in large multimodal models (LMMs) during continual learning. LMMs, which integrate visual and textual processing capabilities, encounter performance degradation when sequentially learning new tasks. To address this, MODALPROMPT leverages dual-modality (image and text) prompt learning to enable continual learning without task-specific expansion or data replay, which can be resource-intensive and raise privacy issues. By combining task-specific prototype prompts with a selection mechanism informed by image-text distributions, the model achieves improved task retention and transfer of knowledge across a variety of multimodal benchmarks.

**Strengths:**

1.	Introduces an innovative, data-efficient solution to catastrophic forgetting, critical for LMM applications in dynamic task environments.
2.	Demonstrates strong empirical performance with improvements across key continual learning metrics.
3.	Efficient design enables lower computational cost, making it scalable for broader application.

**Weaknesses:**

1.	The baseline lacks a comparison with other prompt learning methods.
2.	Complexity in configuring prompt numbers and selection features may limit broader accessibility without further simplification or automation.
3.	ModalPrompt needs to convincingly differentiate itself from prior work in prompt-based continual learning, likely through robust comparative experiments and ablations.

**Questions:**

1.	There are numerous methods for multimodal prompt learning. Did the authors explore other approaches, and if so, how effective were they?
2.	Additionally, why does the baseline comparison only include the LoRA method? Are there other fine-tuning methods considered? Could a direct comparison between LoRA and prompt learning be potentially unfair?
3.	Is there any comparison of FPS, storage, and speed?

---

### Official Review · Reviewer_Nsh2 · 2024-11-02

**Soundness:** 2
**Presentation:** 1
**Contribution:** 2
**Rating:** 3
**Confidence:** 4

**Summary:**

This paper proposes a continual learning scheme for LMMs based on prompt selection and fusion. Experiments on eight datasets show the effectiveness of the proposed method.

**Strengths:**

In large models like LLMs and LMMs, learned prompts serve as new "viewpoints" that enhance the performance of the underlying LMMs on specific tasks. I believe exploring prompt-based "continued learning" techniques can be practically beneficial, especially with the availability of powerful LMMs.

**Weaknesses:**

The paper is difficult to read, as it presents simple ideas in an abstract and complex manner. It requires a substantial revision before one can properly evaluate its soundness and contribution.Thus, I do not believe it is ready for publication at ICLR. Here are some areas of confusion I encountered:
- In line 161, it states “The characteristic of LMM continual learning includes: …” It is unclear whether the authors refer to a general consensus on LMM continual learning or their specific proposals.
- The summation in Eq.(3) lacks a dummy variable. Are you summing over individual prompts within a set for a specific task $t$?
    - Consider using $\bar{x}$ for the average of prompts, as bold symbols could be confusing since they typically represent vectors.
- In line 201, the projection should be defined as $\text{Proj}_v(\cdot):\mathbb{R}^{d_v}\rightarrow\mathbb{R}^{d_t}$.
- In Eq.(7), What is $X_p$? Is it the collection of all prompts? It's unclear how prompts are selected in your process.
    - One possible understanding: You have $N$ prompts for each of the $T$ tasks, so $T\times N$ in total. The selection is performed over all the $T\times N$ and produce $k$ most relevant ones.
- Line 242 states, “To enhance knowledge transfer, the dual-modality features could serve as guiding cues for prompts to accurately get close to multimodal distributions of current task in feature space.” What are the dual-modality features? Are they the features of the current task? What do you mean by “multimodal distributions”? I don't think those terminologies are self-explanatory and commonly used in the field. Why is the closeness to the distribution helpful in enhancing knowledge transfer?
- Eq.(9) abuses the symbol $\mathbf{x}^t_p$ for prototype features,  the same term is used for the “prompt features” in Eq.(3).
- In Eq.(10) what are the definitions of $\alpha^{\le t}$ and $\beta^{\le t}$? What is the shape of $\tilde{X}^t_p$?
- In line 265, where do you define the parameters $\theta_p^t$ of prototype prompts?
- In Table 1, what is the metric of the first two methods?
- In Table 2, what do $B_i$ and $M_i$ represent in the second row?
- Previous text implies that “number of selection prompts $k$” refers to selecting the top-k most similar prompts. However, by line 448-455, it seems $k$ refers to the number of sets of prototype prompts. Which is the correct understanding?
- Line 456 is confusing when it mentions “choosing three sets of prototype prompts.” Based on subsection 3.2 (line 237, “we term set of prompt for each task as prototype prompts”), shouldn’t the number of prototype prompt sets match the number of tasks, which is eight?
- In Fig.5, it is not clear what quantity is plotted. Is it the average similarity between the prototype features and task features across all in task samples and targeting prototypes?

In addition, the visualization subsection at P.10 provides little information. Cherry-picking examples do not represent the overall behavior of your model. and I don't understand how these examples support the claim that your model retains previously learned knowledge.

**Questions:**

- I will ask the authors to revise the entire paper to clarify their method and arguments.
- In the main text the authors repeatedly emphasize that their method is time-efficient in the sense that the time complexity of inference depends on the number of selected prompts rather than tasks. However, I find this unclear. First, during the inference for each task sample, one needs to compute the similarity with all the prompts, whose number equals to the number of tasks. If we disregard such selection computation, why should other methods exhibit an $O(N_{task})$ time complexity?
- To illustrate the importance of the dual-modality guidance, the authors compared the full results with those from using only image or text modalities. This comparison could be biased, as it relies solely on $\alpha$ or $\beta$ for prompts selection in the latter case. To ensure fairness, for example, one could use two different text encoders to obtain two estimates of text-based similarities $\beta$ and $\beta'$. This allows for a comparison of results using  $\alpha + \beta$ with those using $\beta + \beta'$. Can you carry out this comparison and show the results?
- There seems to be a discrepancy between results in Fig.5 and Fig.6: GQA task features show their highest similarity with ImageNet prototype features (Fig. 5). yet the selected prototype prompts are primarily from the GQA task (Fig. 6).

---

### Official Review · Reviewer_roZe · 2024-11-04

**Soundness:** 2
**Presentation:** 3
**Contribution:** 2
**Rating:** 5
**Confidence:** 3

**Summary:**

This paper proposes a dual-modality guided prompt learning framework (ModalPrompt) tailored for multimodal continual learning to effectively leran new tasks while alleviating forgetting of previous knowledge. Extensive experiments demonstrate the superiority of the proposed method.

**Strengths:**

This paper is well written and is the first prompt learning framework for rehearsal-free continual learning of LMMs. The experimental results show a significant improvement, with comparisons conducted across various tasks and datasets.

**Weaknesses:**

1. The proposed method lacks substantial novelty, as prompt learning has already been widely used in fine-tuning pre-trained vision-language models in the continual learning setting.
2. The baseline is too weak, thus the effectiveness of the method is not very convincing. For example, the baseline accuracy of zero-shot on the REC task is 0.00.

**Questions:**

1. Prompt-based continual learning methods like L2P[1], DualPrompt[2], S-Prompts[3] and HiDe-Prompt[4] employ various prompt design and selection strategies. As for the prompt design, how does this paper demonstrate the superiority of the proposed method?
2. Is there a writing error in Equation 12? This loss aims to increase the similarity between $x^t_P$ and $x_{instruct}$; however, as $x^t_P$ and $x_{instruct}$ become more similar, it means the prompt cannot provide additional information, which would be detrimental to prompt learning.

[1] Wang Z, Zhang Z, Lee C Y, et al. Learning to prompt for continual learning[C]//Proceedings of the IEEE/CVF conference on computer vision and pattern recognition. 2022: 139-149.

[2] Wang Z, Zhang Z, Ebrahimi S, et al. Dualprompt: Complementary prompting for rehearsal-free continual learning[C]//European Conference on Computer Vision. Cham: Springer Nature Switzerland, 2022: 631-648.

[3] Wang Y, Huang Z, Hong X. S-prompts learning with pre-trained transformers: An occam’s razor for domain incremental learning[J]. Advances in Neural Information Processing Systems, 2022, 35: 5682-5695.

[4] Wang L, Xie J, Zhang X, et al. Hierarchical decomposition of prompt-based continual learning: Rethinking obscured sub-optimality[J]. Advances in Neural Information Processing Systems, 2024, 36.

---

### Note · Authors · 2024-11-15

I have read and agree with the venue's withdrawal policy on behalf of myself and my co-authors.